# Nutritional Management and Physical Activity in the Treatment of Sarcopenic Obesity: A Review of the Literature

**DOI:** 10.3390/nu16152560

**Published:** 2024-08-04

**Authors:** Yavor Assyov, Iveta Nedeva, Borian Spassov, Antonina Gerganova, Toni Velikov, Zdravko Kamenov, Tsvetelina Velikova

**Affiliations:** 1Department of Internal Medicine, Clinic of Endocrinology, University Hospital “Alexandrovska” Medical University, Georgi Sofiyski 1 Str., 1431 Sofia, Bulgaria; yavovian@abv.bg (Y.A.); antonina_gerganova@yahoo.com (A.G.); zkamenov@hotmail.com (Z.K.); 2Department of Epidemiology and Hygiene, Medical University of Sofia, 1431 Sofia, Bulgaria; iveta_nedeva@yahoo.com; 3Department of Diagnostic Imaging, Medical University, 1431 Sofia, Bulgaria; spassovborian@yahoo.com; 4Clinic of Cardiology, SHATC “Medica Cor” EAD, 7000 Rousse, Bulgaria; toni_velikov@abv.bg; 5Medical Faculty, Sofia University St. Kliment Ohridski, 1407 Sofia, Bulgaria

**Keywords:** obesity, sarcopenia, sarcopenic obesity, resistance training, nutritional management, intervention, elderly

## Abstract

Background: The prevalence of sarcopenic obesity among adults aged ≥65 years is increasing worldwide. It is a condition that describes the concomitant presence of sarcopenia and obesity, but it appears to be associated with greater increases in the risks for disability, morbidity, and mortality than the two conditions combined. The current review aims to summarize the available literature data on the effectiveness of lifestyle modification for the management of this high-risk geriatric syndrome. Methods: We conducted a comprehensive search across multiple databases, including PubMed, Scopus, Web of Science, and Cochrane Library, for publications published from January 1950 to June 2024. Results: The detection of early preventive and therapeutic approaches to combat sarcopenic obesity is essential for healthy aging. There is ample evidence that suggests that poor dietary habits and physical inactivity are the main reasons for the development of sarcopenic obesity and should thus be the main targets for intervention. In the absence of effective pharmacological interventions, the best effect on sarcopenic obesity is achieved by combination with proper dietary intervention and regular physical activity according to the individual’s health condition. Conclusions. Further research is needed to discover the most effective strategy for the prevention and treatment of sarcopenic obesity, as well as potential pharmacological options to improve muscle mass and function in older populations with physical restrictions.

## 1. Introduction

Biological aging is accompanied by a progressive loss of fertility and functional capacity and an increase in mortality. Muscle atrophy, related to age, is the most common type of muscle atrophy, and its importance is increasing significantly due to the demographic development of modern society with a growing proportion of elderly individuals [1]. Exploring the underlying mechanisms of age-related muscle atrophy and hypothesizing potential interventions are crucial for improving the quality of life and functional independence of the aging population.

Hippocrates already spoke of the age-related changes he observed in skeletal musculature, but only in 1989 Irwin Rosenberg suggests the term “sarcopenia” to describe the age-related muscle and strength loss—a problem that has never been described as a separate component of aging. The term “sarcopenia” derives from the Greek word “sarx”—flesh and “penia”—loss [2]. This conceptualization marked a significant advancement in understanding the specific impacts of aging on muscle mass and function, prompting further research into its causes, consequences, and potential treatments.

Sarcopenia is characterized by progressive loss of muscle mass with the advancement of age. The detrimental effects of sarcopenia are reduced muscle strength and the inability to perform normal physical activity, loss of independence, progressive weakness, and a sedentary lifestyle, associated with increased risk of falling and fractures, as well as other adverse health effects [3]. In line with this, early identification and intervention are crucial in managing sarcopenia to mitigate its impact and improve the overall well-being of affected individuals.

According to recent projections, the global population of people over 60 years of age is projected to grow from 600 million in 2000 to over 2 billion by 2050. With the increase in life expectancy and the number of elderly people, sarcopenia has become one of the major problems of public health, and the financial losses to address it are increasing worldwide. It has been estimated that between 5–13% of persons over 60 have sarcopenia, while its prevalence reaches 50% in those over 80 [4].

Obesity has reached epidemic proportions globally, and since 1980, its prevalence has increased more than twofold. According to more recent studies, such as the American NHANES, the prevalence of obesity, standardized by age (BMI > 30 kg/m^2^), is 32.7% [5]. It is also worth noting that obesity is also tightly related to the natural process of aging. It is a well-known fact that a bidirectional relationship exists between visceral fat and muscle mass that impacts the development of the cardio-metabolic syndrome [6]. Furthermore, sarcopenia is related to physical inactivity, which leads to reduced energy expenditure and, subsequently, the development of obesity [7].

Therefore, the term sarcopenic obesity (SO) has been suggested as a condition that describes the concomitant presence of sarcopenia and obesity in an individual [8]. SO has a variety of additive negative health outcomes that impact overall functional ability as well as the overall morbidity and mortality and associated medical costs [9]. Furthermore, SO and aging are characterized by several abnormalities observed in skeletal muscle, such as intramuscular lipid accumulation, which has been suggested to lead to reduced regenerative capacity, resistance to anabolic stimuli such as growth factors, and increased local inflammatory processes [10].

Currently, no approved medications exist for the treatment of SO. That underlines the importance of non-pharmacologic treatment for this condition. The aim of this review is to summarize the available data and the effectiveness of various nutritional and physical interventions for the management of this prevalent disease.

## 2. Methods

For this review, a comprehensive literature search was conducted across multiple databases, including PubMed, Scopus, Web of Science, and Cochrane Library. The search spanned publications from January 1950 to June 2024. Boolean operators were employed to refine the search, using key terms such as (“Sarcopenic Obesity” OR “SO”) AND (“Nutritional Management” OR “Dietary Intervention”) AND (“Physical Activity” OR “Exercise”) AND (“Treatment” OR “Management”) AND (“Clinical Trials” OR “Randomized Controlled Trials” OR “RCTs”). The initial search retrieved a total of 850 articles. After applying inclusion criteria, such as relevance to the topic, studies involving human participants, and publications in English, and excluding duplicates and irrelevant studies, 64 articles were included in the final review (Figure 1). These selected papers provided insights into the role of nutritional management and physical activity in treating SO, highlighting evidence-based strategies and outcomes.

## 3. Nutritional Management of Sarcopenic Obesity

Diet is a significant contributor to the development of both sarcopenia and obesity, as well as SO as an isolated entity. In sarcopenic subjects, nutrition mainly influences the condition via inadequate protein intake. Conversely, in individuals with obesity, we observe excess fat accumulation due to an imbalance between energy consumption and expenditure [11]. In SO, a healthy diet should aim to provide an optimal nutritional intake to counteract skeletal muscle mass loss and/or stimulate its increase and facilitate the reduction of excess fat stores [12]. Unfortunately, in older SO adults, weight loss is not just at the expense of fat mass but also skeletal muscle loss.

SO is a condition that remains controversial regarding its definition, impact on health, and treatment approaches [13]. Studies dedicated to SO prevention and treatment are highly heterogeneous regarding diagnostic criteria and study design. Frequently, physical and nutritional strategies are merged as a sole intervention, making it difficult to assess their individual contribution.

One of the earliest studies investigating the impact of nutrition on body composition in women with SO was that of Aubertin-Leheudre et al. (2007) [14]. Authors established that a six-month isoflavone supplement enhanced fat-free mass [14]. Later, Aleman-Mateo et al. (2012) did not prove that adding 210 g of ricotta cheese daily for three months to a regular diet improves skeletal muscle mass or strength in sarcopenic people with a mean BMI of 26.3 ± 3.8 kg/m^2^ [15]. However, Coker et al. (2012) proved that in elderly individuals, adding whey protein and essential amino acids promotes adipose tissue reduction and benefits to muscle metabolism in a restrictive diet (800 kcal/day) [16]. Other authors also confirm that whey protein supplementation preserves postprandial myofibrillar protein synthesis in older individuals [17,18]. This effect could be ascribed to its high leucine content, which Wall et al. (2013) proved. They found that adding 2.0–2.5 g daily leucine enhances muscle protein formation [19]. Additionally, Muscariello et al. (2016) found that a high protein (1.2 g/kg) hypocaloric diet for three months leads to weight reduction while preserving muscle mass in women with SO [20]. Sammarco et al. (2017) also showed the benefits of this approach [21]. They included women with SO who received placebos or 1.2–1.4 g/kg protein daily. The individuals in the placebo group lost more lean mass [21]. On the other hand, isolated hypocaloric approaches should be avoided in individuals with SO because it has been established that adipose tissue reduction is accompanied by skeletal muscle mass loss [12,22,23,24]. Muscariello et al. (2016) proved that diet regimens low in both calories and protein negatively influence muscle index [20]. All the factors mentioned above are highly undesirable due to the detrimental impact on bone mineral density and optimal microelement levels [25,26].

Another strategy integrated by Kemmler et al. (2017) was whole-body electromyostimulation along with protein supplementation or isolated protein supplementation [27]. They observed 100 Bavarian men and confirmed a significantly favorable effect compared to the non-intervention group [27]. Reduction in body fat and improvement in skeletal muscle index were found in both groups [27]. A meta-analysis by Hsu et al. (2019) aimed to assess the impact of exercise or nutrition on body composition and physical capabilities and establish whether a combined approach or nutrition or exercise alone contributes more to the determinants mentioned above [28]. They included more than a dozen studies and found that nutritional intervention reduced fat mass without improving muscle mass compared to healthy volunteers. The meta-analysis concluded that nutrition alone did not improve grip strength used to determine muscle strength [28]. Moreover, the authors indicated that adding protein-rich products did not provide additional benefits to body composition and metabolic profile [28]. The lack of consistency of results between studies could be attributed to different SO definitions as well as the heterogeneity of studies.

In summary, the majority of studies conclude that elderly people with obesity (>65 years) with sarcopenia should maintain a hypocaloric diet while receiving a higher protein intake of above 1–1.2 g/kg body weight in order to maintain and even improve muscle quantity, quality, and function [12,29,30,31].

Some of the studies on nutritional management of sarcopenia are presented in Table 1.

## 4. Physical Activity and Sarcopenic Obesity

Physical exercise is currently the best and most commonly used tool used to combat SO in the elderly, as evidenced by multiple meta-analyses [28,32]. Unsurprisingly, SO patients are generally less active compared to other individuals with obesity but without sarcopenia, and individuals who exercised more than three times per week were significantly less likely to experience SO traits when compared to individuals not training regularly [33].

### 4.1. Resistance Training

It has been widely documented that resistance training (RT) can increase muscular strength in the elderly [34]. Different exercise regimens, especially ones containing a resistance training component, have also been shown to prevent the development of SO in patients at high risk for it. In combating SO, resistance training seems to be the most effective way to treat SO and multiple exercise RCTs have been carried out using free weights, machines, elastic bands, and sandbags [35,36].

A couple of RCTs examined the use of elastic bands to improve primary and secondary outcomes in SO elderly individuals [37,38,39,40,41]. They mostly used moderate-intensity protocols (13 on the Borg Scale), which roughly equates to 65% of their 1 Repetition Maximum (1RM), as elderly individuals have difficulties training more intensely [42]. Elastic bands have the benefit of being safe and carry a low risk of injury, which makes them suitable for elderly individuals [37]. In one RCT involving 52 women, a 12-week moderate intensity protocol with a progressive increase in intensity showed significant improvements in the exercise group both in body composition and physical capacity, namely substantially increasing muscle quality, FFM, a reduction in total fat mass, and improvement of gait speed. At the end of the trial, a portion of the subjects were no longer sarcopenic [38]. A newer RCT by the same group revealed that besides improved functional mobility and muscle quality, the exercise group, interestingly, also showed a rise in ALM, which was also raised compared to baseline at the 3- and 9-month follow-ups [39]. In contrast, using a similar 12-week protocol, Huang SW et al. found no significant improvements in lean muscle mass (LMM) in the exercise group following the intervention. They showed, however, significant improvements in total fat mass, fat percentage, and bone mass density scores [37]. Similarly, Lee et al., besides improved physical capacity and bone mass density scores, documented no significant changes in body composition (including lean mass and fat mass) [40]. Finally, a RCT by Park et al. of 24 weeks combining walking activities and elastic band resistance exercises five times per week showed amelioration in body composition and physical function as witnessed by a decrease in their body fat percentage and progression in the chair stand-up test as well as an appreciable increase of their ALM and grip strength [41].

Despite the differences in length and intensity, at the bare minimum, resistance training programmes have led to variable increases in strength as opposed to increases in SMM. Exploring the effect of RT’s effect on the SO index, Gadelha et al. devised a three-times-per-week progressive resistance training program for a total of 24 weeks [43]. Subjects in the exercise group demonstrated a significant increase in their FFM but no reduction in their fat mass. Only in the span of 8 weeks with only two sessions per week, the resistance training group in this RCT had increased their SMM, appendicular skeletal muscle mass (ASM), and grip strength compared to baseline, and additionally, subjects also increased their IGF-1 concentrations [44]. In another study using a low-intensity 12-week exercise protocol, subjects decreased their total fat mass and increased their grip strength despite participants only using light 2–5 lbs. sandbags during the sessions [45].

In contrast, subjects involved in a progressive resistance exercise program with a high-speed component neither improved their muscle performance nor their mobility [46]. It must be noted that in studies with similar protocols, various limitations should be noted—such as overall low intensity, a short duration protocol with short sessions, a short protocol (10 weeks) with short sessions, and finally, subjects only had 30 s of rest between resistance exercise sets, which has been documented to hinder strength gain when total sets remain the same [47,48]. Another study employing a 15-week full-body workout programme yielded surprising results. Subjects in the study were split into a high-speed circuit training (HSC) and a strength/hypertrophy (SH) group. At the end of the protocol, subjects in the HSC group showed better Short Performance Physical Battery (SPPB) scores compared to the SH group; however, both groups showed similar improvements in strength, SMI, and reduction of body fat [49].

There is a possibility that response to resistance training might be diminished in elderly individuals with SO, especially when it comes to body composition changes when compared to subjects without SO, as demonstrated by Silva et al. in their 16-week trial. The eight women in the SO cohort showed no improvements in their body composition, whereas the non-SO group had decreased their fat mass and waist-to-hip ratio and increased their FFM. The non-SO group also increased their 1RM to a greater degree compared to the SO group. It must be, however, noted that the SO group consisted of only eight subjects, and the protocol was suboptimal for strength gain as subjects were instructed to perform each repetition for a total of 4 s [50]. If stronger evidence emerges that SO patients have a smaller stimulus to training, protocols should be more rigorously selected to reflect the primary and secondary outcomes. Similar results could be seen in the same-length program by Stoever K et al., where subjects in both the SO and non-SO groups did not show significant improvements in their body composition, represented by their BMI and SMI. The SO group, however, managed to enhance their SPPB score and Physical Performance Test as well as increase their hand-grip strength [51].

### 4.2. Aerobic Training

Aerobic Training (AT) offers the advantage that individuals can choose from a wider range of activities based on their personal preferences [52], which might play a vital role in longer-term adherence. It has been previously demonstrated that aerobic training can promote muscle protein synthesis regardless of age [53]. Aerobic exercise reaching 70 to 80% heart-rate maximum has been shown to reduce body weight and BMI but leads to no changes in FFM and leg muscle size in elderly participants [53]. Conversely, another study involving elderly women with obesity has shown that a 12-week aerobic circuit-based training regimen leads to a reduction in fat mass and an increase in lean mass [54]. Finally, in a study by Chen et al., the aerobic group improved body fat mass and SMM compared to the control group by performing a combination of dance steps twice per week for eight weeks [44]. It should be noted that the literature is significantly scarcer in publications assessing the efficacy of AT on clinical outcomes of sarcopenia compared to resistance training.

### 4.3. Combination Training

Combination training (CT) is another option to combat SO with physical activity. It consists of combining aerobic and resistance training within the same session or performing both types of exercises within the same week. This type of training has been shown to even prevent SO in high-risk individuals such as breast cancer patients following chemotherapy. The combination of self-selected aerobic exercises and a progressive resistance training program involving the lower limbs proved to be effective for the amelioration of body composition as subjects decreased their body weight and fat mass and increased their lean muscle mass when compared to baseline [36]. Another 16-week combination training program, involving twice-per-week resistance training sessions and once-per-week aerobic activities, managed to improve lean muscle mass and metabolic variables as well as decrease fat mass, body weight, and circulating biomarkers when compared to baseline [55]. A longer study of 24 weeks by Park et al. found significant improvements in body composition and physical function as grip strength; additionally, ASM increased, and body fat percentage and waist circumference decreased. The aerobic activities, although frequent (5 times per week), consisted only of various walking activities, which likely mainly contributed to an increase in energy expenditure. It is, however, interesting that even this lower-effort protocol managed to improve outcomes for patients [41]. In another study that used a moderate-intensity combination training protocol, the resistant training component, albeit progressive, used no weights at the beginning and only progressed to 750 g near the end of the study. Despite this, subjects in both groups increased their strength compared to baseline 5 times as well as improved their gait speed and time for up-and-go [56].

Finally, a study by Chen et al. aimed to compare the effectiveness of AT, RT, and CT in subjects with SO in an 8-week time period. At week 8, the SMM, body fat mass, and visceral fat area (VFA) of the RT, AT, and CT groups were far greater than those of the control group. However, the RT group showed the highest grip strength increase, higher knee extensor performance, and greater IGF-1 concentrations compared to the other groups at week 8 [44].

The most important studies on physical exercising on the SO are presented in Table 2.

## 5. Combined Nutritional and Physical Intervention

It is important to highlight that there is a growing body of literature that highlights the benefits of combined nutritional and physical approaches. Maltais et al. (2016) performed a pilot study that investigated the effect of protein supplementation in the form of dairy shakes (13.53 g protein, 7 g essential amino acids) and resistance training [57]. It showed that significant fat mass reduction was present only in the dairy supplement group [57]. Kim et al. (2016), in 139 SO Japanese women, found that exercise + nutrition or nutrition alone did not improve muscle mass and muscle strength and performance. However, it is worth mentioning that their definition of obesity was 32% body fat mass [58]. Trouwborst et al. (2018) reviewed the integrated approach in the prevention and/or treatment of SO [12]. They concluded that a moderate weight reduction dietary regimen, along with physical activity and a high amount of animal protein distributed across the day, have a substantial positive effect on SO parameters [12]. Nabuco et al. (2019) reported results from a randomized controlled trial that examined the effect of whey protein as an add-on therapy to resistance training on body composition and metabolic profile in elderly women [59]. Their research indicated that individuals in the whey protein group increased their appendicular lean mass and decreased total and trunk fat mass [59]. Some studies also suggest that the increase in protein intake during a low-calorie regimen along with exercise could counterfeit the proportion of muscle mass lost along with weight loss [60,61].

In Figure 2, we present the main aspects of the pathophysiology of SO and how nutrition and physical activity could be used to impact these pathological mechanisms.

The systematic review and meta-analysis by Eglseer et al. focused on the effectiveness of nutritional and exercise interventions in elderly people with SO (retirement age). They confirmed that RT led to significantly reduced body fat and increased muscle mass and muscle strength. Furthermore, adding protein to the diet combined with exercising significantly reduced fat mass, supporting the available data [62].

Another umbrella review of meta-analyses of randomized controlled trials by Reiter et al. revealed that nutrition and exercise interventions on people with SO confirmed their positive health effects. However, authors declared that exercise (i.e., resistance, aerobic, mixed) and a hypocaloric diet enriched with proteins did not show a significant effect on selected results for persons with SO than no intervention, although it could be due to small numbers of studies [63].

Finally, SO research perspectives, outlined by the SO global leadership initiative (SOGLI), released an expert consensus on SO, where along with pathophysiology, screening, diagnosis, and staging, discussed treatments of the condition, including nutritional management and physical exercising [64].

Our review on nutritional management and physical activity in the treatment of SO offers several strengths. It provides a comprehensive synthesis of recent literature, covering a broad timeframe and including diverse studies, which enhances the robustness and generalizability of the findings. The use of multiple databases and rigorous selection criteria ensured the inclusion of high-quality, relevant studies, thereby offering a well-rounded perspective on the topic. Furthermore, the review highlights practical implications and evidence-based strategies that can be directly applied to clinical practice, making it highly relevant for healthcare professionals. However, the review also has limitations. The reliance on published studies may introduce publication bias, as studies with negative results are less likely to be published. Additionally, the exclusion of non-English publications could limit the global applicability of the findings. Variations in study design, population characteristics, and intervention methods across the included studies may also introduce heterogeneity, complicating direct comparisons of results. Lastly, the rapidly evolving nature of research in this field means that new findings may emerge that were not captured within the search timeframe. Despite these limitations, the review provides valuable insights and a solid foundation for future research and clinical practice.

## 6. Conclusions and Perspectives

It is estimated that more than one-fifth of the elderly population suffer from SO worldwide. It is characterized by the reduction of skeletal muscle mass, low muscle function, and body fat excess. As a significant health and public concern, it is associated with frailty, disability, and increased morbidity and mortality. In the absence of effective pharmacological interventions, the best effect on SO is achieved by combination with proper dietary intervention and regular physical activity according to the individual’s health condition. However, further research is needed to discover the most effective strategy for the prevention and treatment of SO, as well as potential pharmacological options to improve muscle mass and function in older populations with physical restrictions.

## Figures and Tables

**Figure 1 nutrients-16-02560-f001:**
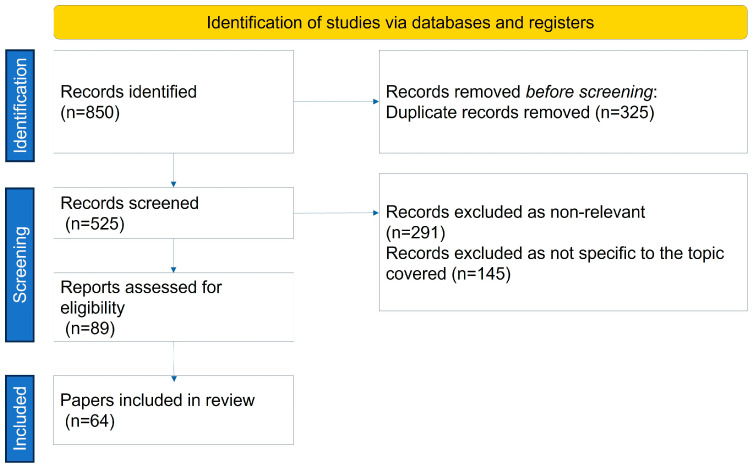
Identification, screening and selection of papers to include.

**Figure 2 nutrients-16-02560-f002:**
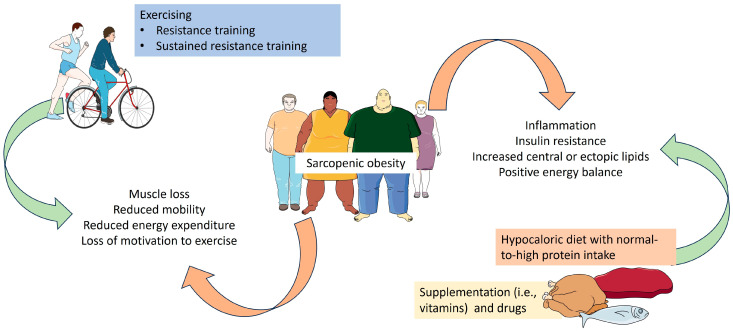
Main aspects of pathophysiology and management of sarcopenic obesity. Parts of the figure were drawn by using pictures from Servier Medical Art. Servier Medical Art by Servier is licensed under a Creative Commons Attribution 3.0 Unported License (https://creativecommons.org/licenses/by/3.0/ Accessed on 1 August 2024).

**Table 1 nutrients-16-02560-t001:** Studies on nutritional management of sarcopenia.

Publication Title	Authors	Length of Intervention	No. of Subjects	Intervention	Clinical Outcome
Six months of isoflavone supplement increases fat-free mass in obese–sarcopenic postmenopausal women: a randomized double-blind controlled trial	Aubertin-Leheudre et al. [14]	6 months	18 postmenopausal women with SO	isoflavone supplementation vs. placebo	- Significantly increased appendicular, leg, and muscle mass index in the isoflavone group
Physiological effects beyond the significant gain in muscle mass in sarcopenic elderly men: evidence from a randomized clinical trial using a protein-rich food	Aleman-Mateo et al. [15]	3 months	40 elderly men and women with sarcopenia over 60 years	Addition of protein-rich food to the diet—210 g/day of ricotta cheese plus the habitual diet vs. habitual diet alone	- total appendicular skeletal muscle (TASM) was not changed significantly, although men in the intervention group gained 270 g in TASM and improved fasting insulin levels - improvement of muscle strength in the intervention group
Whey protein and essential amino acids promote the reduction of adipose tissue and increased muscle protein synthesis during caloric restriction-induced weight loss in elderly, obese individuals	Coker et al. [16]	8 weeks	12 elderly individuals	Caloric restriction-based with meal replacements (EAAMR) vs. competitive meal replacement (CMR) with 400 kcal of solid food	- EAAMR did not promote the preservation of lean tissue - Greater reduction in adipose tissue in EAAMR compared to CMR, with an increase in acute skeletal muscle protein
Whey Protein Supplementation Preserves Postprandial Myofibrillar Protein Synthesis during Short-Term Energy Restriction in Overweight and Obese Adults	Hector et al. [17]	14 days	19 men and 21 women with BMI 28–50 kg/m^2^	Controlled hypocaloric diet (−750 kcal/d)—isolated whey (27 g/supplement) or soy (26 g/supplement) vs. isoenergetic carbohydrate (25 g maltodextrin/supplement)	- Myofibrillar protein synthesis (MPS) was stimulated more when on diets with whey than soy or carbohydrates. - Postprandial MPS was reduced by 9% in the whey group and less in the other groups - Lipolysis was suppressed more with the ingestion of carbohydrates than with soy or whey
Whey protein stimulates postprandial muscle protein accretion more effectively than casein and casein hydrolysate in older men	Pennings et al. [18]	N/A	48 older men aged 74 years	Ingestion of meal-like amount (20 g) of whey, casein, or casein hydrolysate	- Whey protein stimulates postprandial muscle protein accretion more effectively than casein and casein hydrolysate
Leucine co-ingestion improves postprandial muscle protein accretion in elderly men	Wall et al. [19]	N/A	24 elderly men at 74 years	Ingestion of 20 g intrinsically casein protein with (PRO + LEU) or without (PRO) 2.5 g crystalline leucine	- Leucine co-ingestion with a bolus of pure dietary protein further stimulates postprandial muscle protein synthesis
Dietary protein intake in sarcopenic obese older women	Muscariello et al. [20]	3 months	1030 females over 65 years and BMI > 30 kg/m^2^, 104 with sarcopenia	hypocaloric diet (0.8 g/kg desirable body weight/day of proteins) (n = 50), vs. hypocaloric diet with high protein intake (n = 54)	- Protein-rich diets preserve muscle mass in women with sarcopenia
Evaluation of Hypocaloric Diet With Protein Supplementation in Middle-Aged Sarcopenic Obese Women: A Pilot Study	Sammarco et al. [21]	4 months	18 women with obesity aged 41–74 years	Hypocaloric diet plus placebo vs. hypocaloric high-protein diet	- Significantly decreased weight loss in both groups - High-protein diets were associated with the preservation of lean body mass compared to low-calorie diets and improved muscle strength - High-protein diets were associated with improved general health
Function With Enhanced Protein Intake per Meal: A Pilot Study of Weight Reduction in Frail, Obese Older Adults	Porter Starr et al. [23]	6 months	67 (body mass index ≥30 kg/m^2^) older (≥60 years) adults with obesity and a Short Physical Performance Battery score of 4–10	traditional weight loss regimen vs. higher protein intake (>30 g)	- Weight loss in both groups significantly, - Improved function in both groups; greater increase in the protein-diet group
**Micronutrient deficiency in obese subjects undergoing low-calorie diet**	Damms-Machado et al. [25]	3 months	104 subjects	Dietetic intervention with formula diet	- More subjects had micronutrient deficiencies of vitamins (i.e., C), zinc, and lycopene. - A negative correlation between lipophilic serum vitamins and iron and C-reactive protein concentrations and body fat
**Whole-body electromyostimulation and protein supplementation favorably affect sarcopenic obesity in community-dwelling older men at risk: the randomized controlled FranSO study**	Kemmler et al. [27]	16 weeks	100 community-dwelling northern Bavarian men aged ≥70 years with sarcopenia and obesity	Whole-body electromyostimulation with protein supplementation vs. Isolated protein supplementation vs. non-intervention control group	- Significantly favourable effect of first and second group compared to the controls: lost body fat - Skeletal muscle mass increased significantly in the first and second groups - Hand-grip strength increased in the first group

**Table 2 nutrients-16-02560-t002:** Studies on exercising as an intervention for sarcopenic obesity management.

Publication	Authors	Length	No. of Sub.	Intervention	Clinical Outcome
Effects of elastic resistance exercise on body composition and physical capacity in older women with sarcopenic obesity	Liao et al. [38]	12 weeks	46	RT (EB)	- Increased FFM, MQ, and physical capacity (*p* < 0.05) - Fewer patients with sarcopenia (*p* < 0.05) - Fewer patients having physical difficulty (*p* < 0.001)
Effects of elastic band exercise on lean mass and physical capacity in older women with sarcopenic obesity: A randomized controlled trial	Liao et al. [39]	12 weeks	56	RT (EB)	- Significant improvement in ALM and AMI in both follow-ups (all *p* < 0.05)
Body composition influenced by progressive elastic band resistance exercise of sarcopenic obesity elderly women: a pilot randomized controlled trial	Huang et al. [37]	12 weeks	35	RT (EB)	- Body composition improved in both extremities (*p* < 0.04) - Reduction in total fat (*p* = 0.035) and fat percentage (*p* = 0.012) - Improved BMD, T-Score, and Z-Score
Effects of progressive elastic band resistance exercise for aged osteosarcopenic adiposity women	Lee et al. [40]	12 weeks	27	RT (EB)	- no significant differences between groups in TSM (*p* = 0.474), LMI (*p* = 0.330), SMI (*p* = 0.838) and BF% (*p* = 0.148) following the intervention - BMD and T-score (spine) significantly increased (*p* = 0.022 and 0.026)
Effects of 24-Week Aerobic and Resistance Training on Carotid Artery Intima-Media Thickness and Flow Velocity in Elderly Women with Sarcopenic Obesity	Park et al. [41]	24 weeks	50	CT	- Significant reduction in CIMT (*p* < 0.01) and a significant increase in systolic and diastolic flow velocity (*p* < 0.01 and *p* < 0.001), and wall shear rate (*p* < 0.05)
Effects of resistance training on sarcopenic obesity index in older women: A randomized controlled trial	Gadelha et al. [43]	24 weeks	113	RT	- Significant increase in FFM (*p* < 0.01), appendicular FFM (*p* ≤ 0.01), and unchanged Fat Mass, - Improved SO index (*p* < 0.01)
Effects of Different Types of Exercise on Body Composition, Muscle Strength, and IGF-1 in the Elderly with Sarcopenic Obesity	Chen et al. [44]	8 weeks	60	RT, AT, CT	- RT, AT, and CT increased SMM and ASM and reduced BFM and VFA (*p* < 0.05) - RT showed the greatest strength increase in all follow-ups (*p* < 0.05)
Effects of resistance training on body composition and functional capacity among sarcopenic obese residents in long-term care facilities: a preliminary study	Chiu et al. [45]	13 weeks	64	RT	- Significant increase in grip (*p* = 0.001) and pitch strength (*p* = 0.014) - No difference in body fat and SMM (*p* > 0.05)
Effects of a progressive resistance exercise program with high-speed component on the physical function of older women with sarcopenic obesity: a randomized controlled trial	Vasconcelos et al. [46]	10 weeks	28	RT (with high-speed component)	- Insignificant differences in anthropometric measurements and physical function - Increase in knee extensor power (*p* < 0.01)
High-speed circuit training vs. hypertrophy training to improve physical function in sarcopenic obese adults: a randomized controlled trial	Balachandran et al. [49]	15 weeks	21	RT, AT	- Improved SPPB score (HSC > SH, *p* = 0.08) - Increase in leg press power (HCS *p* < 0.01; SH *p* = 0.03) lower body (SH *p* < 0.01) and upper body strength (HCS *p* < 0.01, SH *p* = 0.03)
Resistance training-induced gains in muscle strength, body composition, and functional capacity are attenuated in elderly women with sarcopenic obesity	Silva et al. [50]	16 weeks	49 totalSO (8)Non-SO (41)	RT	- only the non-SO group had significant reductions in BF% (*p* = 0.006), waist circumference (*p* = 0.01), waist-to-hip ratio (*p* = 0.02), and neck circumference (*p* = 0.03) - only the non-SO group increase their 30-s chair stand test performance (*p* = 0.000) and TUG (*p* = 0.000) - both SO (*p* = 0.01) and non-SO (*p* = 0.000) groups gained significant leg press strength
Influences of Resistance Training on Physical Function in Older, Obese Men and Women With Sarcopenia	Stoever et al. [51]	16 weeks	SAR (28)NSAR (20)	RT	- SAR group increased hand-grip strength (*p* = 0.01), while NSAR maintained theirs - SAR group increased gait speed significantly compared to NSAR - NSAR group decreased body weight significantly (*p* = 0.03) compared to SAR - Both groups improved their SPPB and PPT scores (*p* < 0.05)
Effects of Aerobic and Resistance Exercise on Metabolic Syndrome, Sarcopenic Obesity, and Circulating Biomarkers in Overweight or Obese Survivors of Breast Cancer: A Randomized Controlled Trial	Dieli-Conwright et al. [55]	16 weeks	100	CT	- Significant decrease in body weight, BMI (*p* ≤ 0.01) - Significant decrease in appendicular skeletal muscle index and biomarkers for SO (*p* ≤ 0.01) - Significant increase in lean mass (*p* ≤ 0.01)

FFM = Fat-Free Mass; MQ = Muscle Quality; ALM = Appendicular Lean Mass; AMI = Appendicular Muscle Mass; BMD = Bone Mineral Density; TSM = Total Skeletal Muscle Mass; LMI = Lean Muscle Mass Index; SMI = Skeletal Muscle Mass Index; BF% = Body Fat Percentage; CIMT = Carotid Intima-media Thickness; SMM = Skeletal Muscle Mass; BFM = Body Fat Mass; VFA = Visceral Fat Area; SPPB = Short Physical Performance Batter; SH = strength/hypertrophy group; HSC = high-speed circuit group; TUG = Time Up and Go test; SAR = Sarcopenia Group; NSAR = Presarcopenia Group.

## Data Availability

Not applicable.

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
