# Peer review of "Nutritional Management and Physical Activity in the Treatment of Sarcopenic Obesity: A Review of the Literature"

_nutrients, 2024, doi:10.3390/nu16152560_

Round 1
Reviewer 1 Report
Comments and Suggestions for Authors
This is an interesting and well written review article with adequate novelty. Some minor points should be addressed.
- The authors could add subheading in their Abstract (e.g. Background, Methods, Results Conclusions) in order to be more easily readable.
- The authors should include a Methods section in the abstract to provide a brief detail of the methodology used to collect the data used in their review article.
- A flow chart diagram for studies enrollment could also be useful.
- The 1st, 2nd and 3rd paragraphs of the Introduction section need a bit more analysis.
- At the end of the discussion section and a separate paragraph reporting the strengths and the limitations of the present review article should be included.
Comments on the Quality of English LanguageMinor editing of English language required
Author Response
Comments and Suggestions for Authors
This is an interesting and well written review article with adequate novelty. Some minor points should be addressed.
- We thank the reviewer for their constructive feedback, which has significantly contributed to the improvement of our manuscript.
- The authors could add subheading in their Abstract (e.g. Background, Methods, Results Conclusions) in order to be more easily readable.
- We have revised the abstract to include subheadings (Background, Methods, Results, Conclusions) to enhance readability and structure, although the paper is review and not an original article.
- The authors should include a Methods section in the abstract to provide a brief detail of the methodology used to collect the data used in their review article.
- We agree with the referee and added a brief Methods section in the abstract to outline the methodology used in gathering and selecting papers for the review to enhance the transparency of our research process.
- A flow chart diagram for studies enrollment could also be useful.
- A flow chart diagram illustrating the study enrollment process has been included. This visual representation aids in understanding the selection criteria and the flow of studies included in the review, providing readers with a clear overview of our methodology.
- The 1st, 2nd and 3rd paragraphs of the Introduction section need a bit more analysis.
- We expanded the analysis in the 1st, 2nd, and 3rd paragraphs of the Introduction section to provide deeper context and background. This enhancement clarifies the rationale behind the review and sets a more robust foundation for the subsequent discussions.
- At the end of the discussion section and a separate paragraph reporting the strengths and the limitations of the present review article should be included.
- A separate paragraph discussing the strengths and limitations of our review has been added at the end of the Discussion section. This addition provides a balanced view of our findings and acknowledges the scope and constraints of our work.
Comments on the Quality of English Language
Minor editing of English language required
Minor edits have been made throughout the manuscript to refine the language and improve clarity. We appreciate the feedback on this matter and have ensured that the writing meets a high standard of English.
Reviewer 2 Report
Comments and Suggestions for Authors
This is an interesting and well-written article. However, some language correction is needed, e.g. - ” Researches dedicated to SO ….” – “studies” or “research” seem to be a better choice
-“…They included SO women …: - patient-oriented wording should be used, it means women with SO (as we should use “women with obesity” and avoid “obese women”)
- “They included SO women, who receive either placebo” – should be “who received”
Please, go carefully through the text.
Figure 1 - The role of hormone replacement therapy is beyond the scope of this publication, and its use has significant limitations and implications. There is a lack of scientific evidence of its efficacy in SO. Therefore, it isn't easy to understand why the authors emphasized this therapeutic action.
Comments on the Quality of English Languageminor editing required
Author Response
Comments and Suggestions for Authors
- We appreciate the reviewer’s constructive feedback and believe these changes have strengthened the manuscript. Thank you for your insightful suggestions.
This is an interesting and well-written article. However, some language correction is needed, e.g. - ” Researches dedicated to SO ….” – “studies” or “research” seem to be a better choice
-“…They included SO women …: - patient-oriented wording should be used, it means women with SO (as we should use “women with obesity” and avoid “obese women”)
- “They included SO women, who receive either placebo” – should be “who received”
Please, go carefully through the text.
- We are very thankful to the reviewer for pointing to us this. We have revised the manuscript to improve language accuracy, including to update the language to patient-oriented terminology. Throughout the manuscript, we have ensured consistent use of patient-oriented language to describe conditions and populations, therefore, to align our language and style with contemporary best practices in medical writing and patient care communication.
Figure 1 - The role of hormone replacement therapy is beyond the scope of this publication, and its use has significant limitations and implications. There is a lack of scientific evidence of its efficacy in SO. Therefore, it isn't easy to understand why the authors emphasized this therapeutic action.
- We agree with the referee`s arguments and remove HRT from the figure..
Comments on the Quality of English Language
minor editing required
- We have undertaken a comprehensive review of the manuscript for minor language errors and clarity improvements. These revisions aim to enhance readability and precision.